# Corporate Dividend Policies during the COVID-19 Pandemic

Nasir Ali [1], Muhammad Zia Ur Rehman [1], Badar Nadeem Ashraf [2,*] and Falik Shear [1]

1    Faisalabad Business School, National Textile University, Faisalabad 37610, Pakistan
2    LSBU Business School, London South Bank University, London SE1 0AA, UK
*    Correspondence: badarfcma@gmail.com

**Abstract:** In this paper, we examine the changes in corporate dividend policies during the COVID-19 shock. For empirical analysis, we employ annual data of 360 companies from the Pakistan Stock Exchange over the period 2015–2020. Using descriptive analysis and Logit regression models, we find that firms were more likely to either omit or reduce dividend payments during the pandemic year of 2020 as compared to the trends in pre-COVID-19 years of 2015–2019. Further, firms with higher profitability, asset turnover and size were less likely to opt for dividend omissions. On the contrary, dividend omissions were more likely among firms with higher debt ratios. The findings of this study helps to understand firm dividend policies during crisis periods.

**Keywords:** COVID-19; dividend policy; logit regression; Pakistan

## 1. Introduction

The COVID-19 pandemic has wreaked havoc across the globe since its inception in January 2020 in China. As of 3 March 2022, there have been 437.3 million confirmed cases and a total of 5.96 million deaths around the globe (Hale et al. 2021). Together with risks to lives, it has brought grave economic consequences. Economies especially suffered during the year 2020 when we knew little about the spread patterns of the virus and there were no vaccines.

COVID-19-led social distancing policies shut down most of the economic activity and adversely affected the operations of corporate firms. Voluminous literature has shown that stock prices negatively reacted to COVID-19 outbreaks (Alfaro et al. 2020; Ashraf 2020a, 2020b, 2021; Baker et al. 2020; Bavel et al. 2020; Mazur et al. 2021; Phan and Narayan 2020; Ramelli and Wagner 2020). In response, firms not only changed their day-to-day operations, such as policies to work from home, but also their financing and investing decisions including dividend payout policies (Didier et al. 2021; Meyer et al. 2022; Wang et al. 2020).

While troubled firms are likely to omit or cut dividends, signaling theory suggests that firms may maintain or even increase dividend payouts during crisis to signal a better performance to the market (Bhattacharya 1979; Booth and Chang 2011; Caton et al. 2003; John and Williams 1985). Consistent with these theoretical predictions, recent studies have reached mixed conclusions regarding the changes in firm dividend patterns during COVID-19. For instance, Cejnek et al. (2021) and Krieger et al. (2021) found that the number of firms which either omit or cut dividends increased significantly during the COVID-19. On the other hand, Mazur et al. (2021), Ali (2022) and Tinungki et al. (2022) find that most of the firms were able to either maintain or even increase dividends during COVID-19. Given these mixed findings, in this paper we study changes in dividend policies taking a sample of firms from Pakistan.

Pakistan, a south Asian nation, has observed 477,208 total confirmed cases as of 31 December 2020, as depicted in Figure 1. Like other nations, Pakistani markets were hit hard in response to the COVID-19 outbreak. For instance, the KSE-100 index, which is the representative index of the Pakistan Stock Exchange (PSX), declined by around 36 percent

from its highest value of 43,167 on 17 January 2020, to the lowest value of 27,228 on 25 March 2020, suggested a substantial impact on the corporate sector.

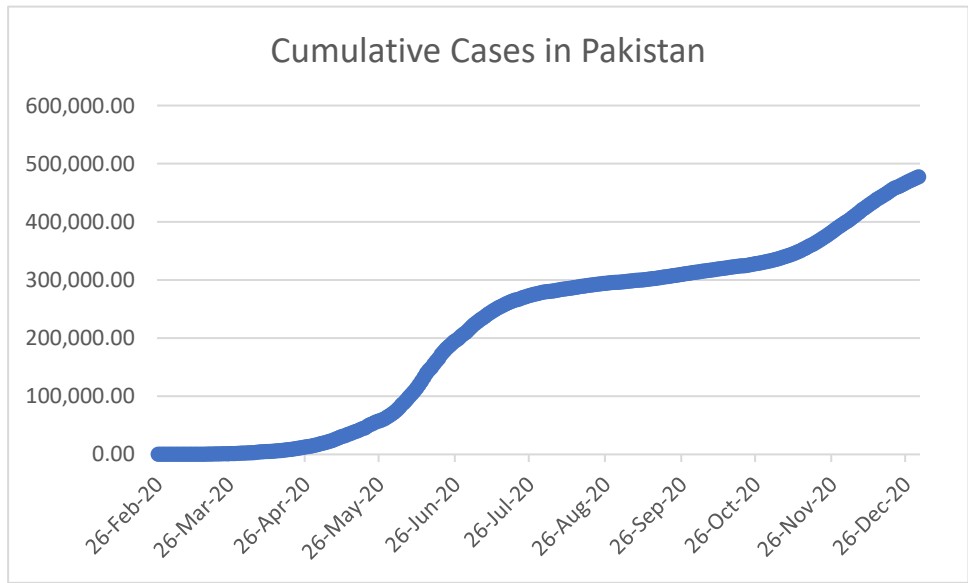

**Figure 1.** Daily COVID-19 cases in Pakistan.

For empirical analysis, we collected annual data of 360 companies from the Pakistan Stock Exchange over the period 2015–2020. Our analysis suggests that firms were significantly more likely to omit or reduce the amounts of dividend payments during the year 2020 when compared to the pre-COVID-19 trends in the sample period. We also observe that firm-level characteristics such as profitability, asset turnover and size affect firms' dividend policies.

We contribute to the extant literature in numerous ways. First and foremost, we complement the studies that shed light on firm dividend policies during the COVID-19 shock (Ali 2022; Cejnek et al. 2021; Krieger et al. 2021; Mazur et al. 2021; Tinungki et al. 2022). These studies have largely used samples from developed nations and report mixed evidence. Extending this debate, we utilize firm-level annual data from Pakistan to test firms' dividend adjustments during the COVID-19 shock.

Our study is related to recent literature which has tested the effects of COVID-19 on firms (Acharya and Steffen 2020; Albuquerque et al. 2020; Ashraf and Goodell 2021; Bae et al. 2021; Baek et al. 2020; Matos et al. 2021; Mirza et al. 2022; Sharif et al. 2020; Shear and Ashraf 2022). Overall, these studies show that the outbreak of COVID-19 and related social distancing policies have had adverse effects on firms irrespective of their business models or other firm level factors. We add to this knowledge by showing that COVID-19 affected firm dividend policies.

Last but not least, we also complement the studies that investigate dividend policies of Pakistani firms (Ghafoor et al. 2014; Khan and Shamim 2017; Khan et al. 2011, 2017; Mirza and Azfa 2010). These studies largely examine how firm-level characteristics, such as size, leverage, profitability, cash holdings and managerial and individual ownership, and country-level factors, such as inflation and taxation, influence firm dividend decisions.

The rest of the paper proceeds as follows: the second part presents a review of the literature on firm dividend policies. The third part covers data and methodology. The fourth part reports the empirical findings. The final part concludes the study.

## 2. Literature Review

Dividends are one of the most commonly debated topics in the finance literature. Different explanations have been offered to explain why firms pay, do not pay or pay higher dividends. Miller and Modigliani (1961) suggested that dividends are irrelevant in

perfect capital markets with zero transaction costs. Black (1976) argued that dividends are a puzzle that needs to be solved. Later research has suggested agency, signaling, life cycle, tax, clientele effects and catering theories.

Dividend payments control agency problems by reducing the cash held by a firm and hence limiting the options for managers to expropriate excess cash or invest in suboptimal projects (Easterbrook 1984; La Porta et al. 2000; Myers 2000; Rozeff 1982).

Dividend payments' initiation or increases provide a positive signal to the market regarding a firms' future profitability, while omitting or cutting dividends provides a negative signal (Bhattacharya 1979; Booth and Chang 2011; Caton et al. 2003; John and Williams 1985).

Life cycle theory suggests that firms pay more dividends in the maturity stage and lower in the growth stage (Brockman and Unlu 2011; DeAngelo et al. 2006; Fama and French 2001).

Tax incentives change dividend demand by the investors; some investors prefer capital gains due to lower tax rates while some prefer dividends to obtain cash for expense. Firms design dividend payments to cater to the preference of investors (Allen et al. 2000; Baker and Wurgler 2004a, 2004b; Foley et al. 2007; Miller and Scholes 1978, 1982; Pettit 1977).

Survey based studies by Baker and his co-authors (Baker and Weigand (2015), Baker and Jabbouri (2016), Baker and Jabbouri (2017) and Baker et al. (2018), among others, have found that both managers and investors agree that catering, bird-in-the-hand, life cycle, signaling and agency theories play substantial role in dividend payment decisions.

Some firm-level factors have also shown a consistent relationship with firm dividend payouts. For example, Fama and French (2001) report that firms which are profitable, growing and larger pay higher dividends. Other studies have shown that country-level factors such as inflation (Basse 2009; Basse and Reddemann 2011), culture (Shao et al. 2009; Zheng and Ashraf 2014) and legal institutions (Ashraf et al. 2016; Ashraf and Zheng 2015) are significant determinants of cross-country differences in dividend policies.

There has been much debate on how firms adjust their dividend policies during crisis periods. The market generally reacts negatively to dividend omissions or cuts and the majority of the managers are hesitant to send a negative signal by reducing dividends. However in times of crisis, cutting or omitting dividends may provide firms with additional cash and flexibility to counter uncertainties.

A scarce number of recent studies have shed light on dividend adjustments by firms during COVID-19. Some of these studies report that on average firms were more likely to either omit or cut dividends. For instance, Krieger et al. (2021) use data of 1400 dividend paying US stocks over the period 2015–2020 and find that the firms were three to five times more likely to cut or omit dividends during the second quarter of 2020 than in any other quarter since the beginning of the sample in 2015. Likewise, Cejnek et al. (2021) report that dividends substantially decreased during the first quarter of 2020 and did not recover until the end of the first year of COVID-19. On the contrary, studies such as Mazur et al. (2021), Ali (2022) and Tinungki et al. (2022) reach conclusions that on average firms were able to either maintain or even increase dividends during COVID-19 and the effect of the pandemic on dividends was not substantial. For instance, Ali (2022) employs a sample of 8889 listed firms from the G-12 countries and supports the signaling hypothesis that most of the sample firms either maintained or increased dividends during the COVID-19 period to avoid a bad signal.

Given these mixed findings, in this paper we study changes in dividend policies taking a sample of firms from Pakistan.

## 3. Data and Methodology

We obtained data of all companies listed in the Pakistan Stock Exchange. In line with Ali (2022), we used data from 2015 to 2020. Our initial sample consists of around 470 companies. However, companies with missing data were omitted from the sample. Thus, our final sample consists of 360 companies, which gives us 2160 firm-year observations.

Data was collected from the State Bank of Pakistan, Pakistan Stock Exchange and investing.com websites. We deleted financial firms from the sample because their financial ratios are not comparable to those of industrial firms (Xu et al. 2021). The sample includes all non-financial companies that are included in the state bank of Pakistan's financial statement analysis (from 2015 to 2020).

Following Ali (2022) and Nissim and Ziv (2001), the dividend change rate is calculated as the percentage difference between company i's dividend in fiscal year t, and the previous fiscal year t−1 as:

$$\Delta \text{DIV}_{i,t} = \frac{\text{DIV}_{i,t} - \text{DIV}_{i,t-1}}{\text{DIV}_{i,t-1}}$$

*Empirical Model*

We use the logit regression model to explore the impact of different firms' specific characteristics on dividend payout decision. Our baseline model is as follows:

$$\text{DIV}_{i,t} = \beta_0 + \beta_1 ROA_{i,t}/CHE_{i,t} + \beta_2 Covid_t + \beta_3 Ast-Tvr_{i,t} + \beta_4 Lev_{i,t} + \beta_5 Liq_{i,t} + \beta_6 Mkt-Bok_{i,t} + \beta_7 Size_{i,t} + \varepsilon_{i,t} \quad (1)$$

where DIV represents divdend-related decisions (as described in Table 1: Panel A) of the firm *i* during year *t*. This is a dummy variable which takes the value of 1 for a particular decision and 0 otherwise. For example, for omission vs increase decision, $\text{DIV}_{i,t}$ is 1 if firm *i* opted for dividend omission (did not pay the dividend) during year *t* and 0 if the firm opted for dividend increase (increased dividend payment). ROA is return on assets, CHE is Change in earnings, Ast-Tvr is asset turnover, Lev is leverage, Liq is liquidity, Size is the natural logarithm of total assets and Mkt-Bok refers to market to book ratio. COVID is a dummy variable which takes the value of 1 during year 2020 and 0 otherwise. Table 1 summarise the variables and their definitions.

**Table 1.** This table presents variables, their definitions, and measurements used in the study.

| Variable | Definition | Measurement |
|---|---|---|
| **Panel A: Dividend-Related Decisions** | | |
| DOM vs. DIC | Omit vs. Increase Dividend | Dummy variable (1 = Dividend Omissions, and 0 for Increase in Dividend) |
| DOM vs. DDC | Omit vs. Decrease Dividends | Dummy variable (1 = Dividend Omissions, and 0 for Decrease in Dividend) |
| DOM vs. DNC | Omit vs. No-Change Dividend | Dummy variable (1 = Dividend Omissions, and 0 for No-Change in Dividend) |
| DIC vs. DDC | Increase vs. Decrease Dividend | Dummy variable (1 = Dividend Increase, and 0 for Decrease in Dividend) |
| DIC vs. DNC | Increase vs. No-Change Dividend | Dummy variable (1 = Dividend Increase, and 0 for No-change in Dividend) |
| DDC vs. DNC | Decrease vs. No-Change Dividend | Dummy variable (1 = Dividend Decrease, and 0 for No-change in Dividend) |
| **Panel B: Firm Specific Control Variables** | | |
| ROA (%) | Return on Assets | $\frac{\text{Net Income}}{\text{Total Assets}}$ |
| CHE (%) | Change in Earnings | $\frac{\text{Net Income}_{i,t} - \text{Net Income}_{i,t-1}}{\text{Net Income}_{i,t-1}}$ |
| Ast-Tvr | Asset Turnover | $\frac{\text{Sales}}{\text{Total Assets}}$ |
| Lev (%) | Leverage | $\frac{\text{Long term Debt}}{\text{Total Assets}}$ |
| Liq | Firm's Liquidity | $\frac{\text{Current Assets}}{\text{Current Liabilities}}$ |
| Mkt-Bok | Market-to-Book | $\frac{\text{Market Value of Equity}}{\text{Book Value of Equity}}$ |
| Size | Firm Size | Natural log of total assets |

## 4. Results and Discussions

### 4.1. Changes in Dividend Policies during the COVID-19

For an overall assessment of firm dividend policies over the sample period, Table 2 presents yearly sample distribution of firms with dividend omissions, decrease, no change and increase. Columns 1 to 4 describe Dividend Omission (DOM), Dividend Decrease (DDC), Dividend No Change (DNC) and Dividend Increase (DIC), respectively. Statistics from column 1 show that there is a mixed pattern of DOM during the sample years. For instance, firms that opted for Dividend omission are the same during the first 2 years. This pattern saw a decline in 2018 and a gradual increase during the last two years. When we

compare the COVID-19 pandemic year (i.e., 2020) to previous years, we observe that there has been a maximum DOM in this year i.e., 62.5%, which is 12% higher from the previous year. This implies that DOM has increased during the pandemic period in sample firms.

**Table 2.** This table reports yearly sample distribution of firm dividend policies.

| Years | Dividend Omission | | Dividend Decrease | | Dividend No Change | | Dividend Increase | | Total |
|---|---|---|---|---|---|---|---|---|---|
| 2016 | 168 | (47%) | 44 | (12%) | 18 | (5%) | 130 | (36%) | 360 |
| 2017 | 170 | (47%) | 57 | (16%) | 18 | (5%) | 115 | (32%) | 360 |
| 2018 | 160 | (44%) | 80 | (22%) | 14 | (4%) | 106 | (29%) | 360 |
| 2019 | 180 | (50%) | 83 | (23%) | 17 | (5%) | 80 | (22%) | 360 |
| 2020 | 225 | (62.5%) | 49 | (13.6%) | 21 | (5.8%) | 65 | (18%) | 360 |

From column 2, we can observe a gradual increase in DDC for sample firms. However, this trend reverses during the COVID-19 year where 13.6% of the companies opted for DDC. This implies that firms avoided decreasing their dividend to avoid negative signals, as these dividends serve as a signal of anticipated financial flows (Bhattacharya 1979). It is evident (column 3) that the highest number of firms (21) maintained their previous dividend payout levels, i.e., they opted for a DNC policy, which implies that they tried to give positive signals to the market. The last column discusses DIC over the sample period. We can observe a gradual decrease in DIC over the years. However, during the pandemic year, only 65 (18%) of the companies (lowest in the sample) increased their dividend to generate positive signals. In a nutshell, our preliminary analysis shows that during the pandemic dividend omission is higher in comparison to previous years and it remained a popular choice among firms during the pandemic year. DIC is the lowest during the pandemic. These findings are in contrast to the findings of Ali (2022), who found that the majority of the firms either maintained or increased their dividend payouts to generate positive signals to the market in G-12 Countries. The findings are in line with the findings of Krieger et al. (2021), who found higher divided cuts during the second quarter of 2020 for U.S. firms.

### 4.2. Drivers of Different Dividend Policies

To obtain insights into different firm characteristics that drive the firms' dividend policies, Table 3 presents a summary of different firm characteristics for DOM, DDC, DNC and DIC groups. The dividend increase group (Panel 4) had better profitability and earnings, compared to all other groups. The ROA in this group, i.e., 11.06%, is much higher than in all other groups. Similarly, this group has the highest asset turnover and size. These statistics imply that higher profitability, ROA, size, and asset turnover are the salient features of the firms that opted for DIC during the sample period.

**Table 3.** Characteristics of dividend change groups.

| Variable | Obs | Mean | Std. Dev. | Min | Max |
|---|---|---|---|---|---|
| Dividend Omission | | | | | |
| ROA % | 910 | −3.675 | 22.051 | −156.677 | 337.917 |
| CHE % | 910 | −1.704 | 23.995 | −430.684 | 117.572 |
| Ast-Tvr | 910 | 0.610 | 0.587 | 0.000 | 3.790 |
| Lev % | 910 | 1.039 | 1.624 | 0.000 | 16.302 |
| Liq | 910 | 2.256 | 14.704 | 0.000 | 316.832 |
| Size | 909 | 14.599 | 2.058 | 8.168 | 20.371 |
| Mkt-Bok | 645 | −10.991 | 622.765 | −15,400.100 | 2421.885 |

**Table 3.** *Cont.*

| Variable | Obs | Mean | Std. Dev. | Min | Max |
|---|---|---|---|---|---|
| Dividend Decrease | | | | | |
| ROA % | 473 | 4.278 | 11.085 | −160.301 | 63.454 |
| CHE % | 473 | −0.714 | 9.943 | −168.600 | 117.572 |
| Ast-Tvr | 473 | 1.129 | 0.710 | 0.000 | 4.821 |
| Lev % | 473 | 0.562 | 0.598 | 0.000 | 12.637 |
| Liq | 473 | 1.793 | 2.277 | 0.000 | 38.361 |
| Size | 473 | 16.382 | 1.850 | 9.744 | 22.057 |
| Mkt-Bok | 344 | −25.026 | 852.857 | −15,400.100 | 2421.885 |
| Dividend No-Change | | | | | |
| ROA % | 88 | 6.661 | 5.692 | −7.055 | 22.834 |
| CHE % | 88 | 0.031 | 1.502 | −10.349 | 4.809 |
| Ast-Tvr | 88 | 1.020 | 0.510 | 0.000 | 2.401 |
| Lev % | 88 | 0.485 | 0.186 | 0.000 | 0.848 |
| Liq | 88 | 1.673 | 0.964 | 1.000 | 5.517 |
| Size | 88 | 16.145 | 1.551 | 12.071 | 19.661 |
| Mkt-Bok | 72 | −25.967 | 211.900 | −1770.900 | 95.662 |
| Dividend Increase | | | | | |
| ROA % | 393 | 11.068 | 9.106 | −5.555 | 57.326 |
| CHE % | 393 | 0.902 | 4.957 | −12.051 | 81.727 |
| Ast-Tvr | 393 | 1.214 | 0.660 | 0.000 | 4.168 |
| Lev % | 393 | 0.501 | 0.208 | 0.000 | 1.252 |
| Liq | 393 | 2.004 | 2.177 | 0.000 | 33.863 |
| Size | 393 | 16.574 | 1.655 | 12.152 | 21.713 |
| Mkt-Bok | 281 | 6.561 | 238.946 | −3604.100 | 1054.633 |

ROA refers to Return on Assets, CHE is Change in earnings, Ast-Tvr Asset is Turn-over, Lev is Leverage, Liq Liquidity, Size natural log of Total Assets and Mkt-Bok refers to Market to Book ratio.

Table 4 presents summary statistics of different dividend policy groups during the pandemic year. It can be noted that firms with the highest (lowest) ROA and liquidity have opted for DIC (DOM) during the pandemic year. Contrarily, high (low) leverage firms have opted for DOM (DIC).

**Table 4.** Characteristics of dividend change groups during COVID-19.

| Variable | Obs | Mean | Std. Dev. | Min | Max |
|---|---|---|---|---|---|
| Dividend Omission | | | | | |
| ROA % | 225 | −1.127 | 34.245 | −71.007 | 337.917 |
| CHE % | 225 | −2.760 | 21.857 | −265.940 | 40.059 |
| Ast-Tvr | 225 | 0.645 | 0.595 | 0.000 | 2.978 |
| Lev % | 225 | 0.953 | 1.529 | 0.000 | 16.302 |
| Liq | 225 | 1.839 | 5.786 | 0.000 | 54.378 |
| Size | 225 | 15.107 | 2.150 | 8.168 | 20.371 |
| Mkt-Bok | 157 | 47.848 | 280.500 | −26.017 | 2421.885 |
| Dividend Decrease | | | | | |
| ROA % | 49 | 4.604 | 5.421 | −7.983 | 18.470 |
| CHE % | 49 | −0.471 | 1.169 | −6.661 | 3.251 |
| Ast-Tvr | 49 | 1.043 | 0.680 | 0.231 | 4.119 |
| Lev % | 49 | 0.500 | 0.218 | 0.096 | 1.206 |
| Liq | 49 | 2.088 | 1.477 | 0.502 | 7.723 |
| Size | 49 | 17.042 | 2.051 | 12.894 | 22.057 |
| Mkt-Bok | 32 | −12.967 | 80.992 | −435.048 | 57.713 |

**Table 4.** *Cont.*

| Variable | Obs | Mean | Std. Dev. | Min | Max |
|---|---|---|---|---|---|
| Dividend No-Change | | | | | |
| ROA % | 21 | 6.327 | 6.520 | −4.925 | 19.289 |
| CHE % | 21 | −0.384 | 0.956 | −3.085 | 0.880 |
| Ast-Tvr | 21 | 0.912 | 0.435 | 0.305 | 1.819 |
| Lev % | 21 | 0.494 | 0.135 | 0.227 | 0.746 |
| Liq | 21 | 1.473 | 0.406 | 0.876 | 2.419 |
| Size | 21 | 16.384 | 1.382 | 12.651 | 18.696 |
| Mkt-Bok | 18 | −5.619 | 55.552 | −224.033 | 43.399 |
| Dividend Increase | | | | | |
| ROA % | 65 | 7.830 | 17.838 | −94.108 | 56.690 |
| CHE % | 65 | 0.258 | 2.369 | −5.223 | 15.312 |
| Ast-Tvr | 65 | 0.963 | 0.615 | 0.000 | 2.239 |
| Lev % | 65 | 0.514 | 0.227 | 0.008 | 1.127 |
| Liq | 65 | 3.393 | 14.352 | 0.155 | 117.016 |
| Size | 65 | 16.161 | 1.996 | 10.032 | 20.810 |
| Mkt-Bok | 47 | 20.327 | 190.132 | −686.593 | 1054.633 |

ROA refers to Return on Assets, CHE is Change in earnings, Ast-Tvr Asset is Turn-over, Lev is Leverage, Liq is Liquidity, Size is natural log of Total Assets and Mkt-Bok refers to Market to Book ratio.

### 4.3. Why Firms Prefer One Dividend Policy over Others?

Our descriptive analysis demonstrates that firm-specific factors have an impact on firms' decisions regarding their dividend policy. To ascertain why firms prefer a specific dividend policy and what is the most preferred choice during the pandemic, we use logit regression models as described in Equation (1). Tables 5 and 6 show the logit regression results for Omission vs. Increase/Decrease/No-Change Dividend, for Increase vs. Decrease/No-change Dividend and Decrease vs. No-Change Dividend, respectively. We have used two alternative measures of profitability, i.e., ROA (Model I, III, VI, VIII and X) and CHE (Model II, IV, V, VII, IX and XI), and other firm-specific characteristics as control variables.

Model I–II present results for Dividend Omission vs. Dividend Increase policy choice. It is evident that ROA, Asset turnover and Size have a negative and significant coefficient, implying that firms with higher values in these characteristics have tried to increase their dividend rather than omitting the dividend payout. The findings provide support to the argument of Ali (2022) that companies with rising dividends are typically larger in size. These findings augment our findings from Table 3. Leverage has a significantly positive coefficient in models I and II, indicating that high risk firms are more likely to omit dividends rather than increasing them. Due to leverage, the firms are at high risk and they are more likely to omit or cease dividends. This phenomena has been supported by previous studies, e.g., Cejnek et al. (2021) note that leverage and the decline in dividend are both positively correlated.

Model III–IV describe results for Dividend Omission vs. Dividend Decrease policy choice. It is evident that firms with higher (lower) levels of Asset turnover, Size and liquidity are going to decrease dividends rather than omitting dividend payout.

An interesting finding is that COVID-19 has a positive and significant coefficient in models I-IV, which implies that firms preferred Dividend Omissions rather than Dividend Increase/Decrease in COVID-19. The finding is in line with our descriptive analysis, but in contrast to Ali (2022) who found that the majority of the firms either maintained or increased their dividend payouts during COVID-19. The finding supports those of Krieger et al. (2021), who found higher divided cuts during the second quarter of 2020 for U.S. firms.

**Table 5.** Determinants of Dividend Payout Policy.

| Variables | Panel A | | Panel B | | Panel C |
|---|---|---|---|---|---|
| | Omit vs. Increase Dividend | Omit vs. Increase Dividend | Omit vs. Decrease Dividends | Omit vs. Decrease Dividends | Omit vs. No-Change Dividends |
| | Model I | Model II | Model III | Model IV | Model V |
| ROA | −0.379 *** | | −0.04 | | |
| | (0.1050) | | (0.0322) | | |
| Ast-Tvr | −3.828 ** | −5.031 *** | −2.323 *** | −2.850 *** | −6.707 |
| | (1.6870) | (1.3320) | (0.7560) | (0.8230) | (5.2270) |
| Lev | 10.18 ** | 10.58 *** | −2.074 | −2.666 | 77.45 |
| | (4.0850) | (3.2780) | (1.7730) | (2.0160) | (52.6800) |
| Liq | 0.249 | −1.098 | −2.487 *** | −3.192 *** | 7.659 |
| | (0.9780) | (0.8440) | (0.6610) | (0.8010) | (5.0670) |
| Size | −2.481 * | −2.151 * | −2.501 *** | −1.564 * | −16.89 |
| | (1.2980) | (1.1630) | (0.7570) | (0.8780) | (11.8800) |
| Mkt-Bok | 0.0294 | 0.00475 | 0.00624 | 0.00853 | 0.000642 |
| | (0.0307) | (0.0133) | (0.0224) | (0.0245) | (0.0204) |
| COVID-19 | 2.131 ** | 2.577 *** | 0.645 * | 0.831 ** | 12.01 |
| | (0.8490) | (0.7420) | (0.3540) | (0.3810) | (13.8800) |
| CHE | | −0.0257 | | −0.000739 | −0.0391 |
| | | (0.0225) | | (0.0109) | (0.0439) |
| Observations | 260 | 232 | 301 | 248 | 68 |
| Number of com | 72 | 69 | 78 | 70 | 24 |
| Firm Fe | Yes | Yes | Yes | Yes | Yes |

ROA refers to Return on Assets, CHE Change in earnings, Ast-Tvr Asset Turn-over, Lev Leverage, Liq Liquidity, Size = natural log of Total Assets, Mkt-Bok Market to Book ratio. COVID-19 is a dummy variable which takes a value of 1 during 2020 and 0 otherwise. ***, ** and * represent significance levels at 1%, 5% and 10% levels, respectively.

**Table 6.** Determinants of Dividend Payout Policy.

| Variables | Panel D | | Panel E | | Panel F | |
|---|---|---|---|---|---|---|
| | Increase vs. Decrease Dividend | Increase vs. Decrease Dividend | Increase vs. No-Change Dividend | Increase vs. No-Change Dividend | Decrease vs. No-Change Dividend | Decrease vs. No-Change Dividend |
| | Model VI | Model VII | Model VIII | Model IX | Model X | Model XI |
| ROA | 0.288 *** | | 0.265 *** | | −0.119 * | |
| | (0.041) | | (0.097) | | (0.072) | |
| Ast-Tvr | −1.593 ** | 1.584 *** | −1.054 | −0.219 | −1.89 | −2.437 * |
| | (0.787) | (0.548) | (1.692) | (1.496) | (1.483) | (1.396) |
| Lev | −1.997 | −2.051 | 1.29 | −0.0921 | 4.373 | 5.854 * |
| | (1.651) | (1.551) | (3.572) | (3.395) | (3.515) | (3.535) |
| Liq | −0.0211 | 0.488 * | 0.0141 | 0.287 | 0.087 | 0.0512 |
| | (0.285) | (0.269) | (0.437) | (0.435) | (0.439) | (0.446) |
| Size | −0.714 | 0.27 | −1.897 | −0.952 | 0.00489 | −0.285 |
| | (0.720) | (0.616) | (1.527) | (1.340) | (1.659) | (1.583) |
| Mkt-Bok | −0.0011 | −0.000967 | 0.000706 | 0.000332 | 0.00354 | 0.0038 |
| | (0.002) | (0.001) | (0.001) | (0.001) | (0.003) | (0.003) |
| COVID-19 | −0.468 | −0.731 ** | −0.512 | −0.803 | −0.88 | −0.782 |
| | (0.371) | (0.308) | (0.606) | (0.562) | (0.597) | (0.572) |
| CHE | | 0.0356 * | | 0.191 | | −0.260 * |
| | | (0.018) | | (0.179) | | (0.155) |
| Observations | 509 | 509 | 150 | 150 | 145 | 145 |
| Number of com | 128 | 128 | 46 | 46 | 49 | 49 |
| Firm Fe | Yes | Yes | Yes | Yes | Yes | Yes |

Where ROA refers to Return on Assets, CHE Change in earnings, Ast-Tvr Asset Turn-over, Lev Leverage, Liq Liquidity, Size natural log of Total Assets, Mkt-Bok Market to Book ratio. COVID-19 is a dummy variable which takes a value of 1 during 2020 and 0 otherwise. ***, ** and * represent significance levels at 1%, 5% and 10% levels, respectively.

Model VI–VII (Table 6) present results for Dividend Increase vs. Decrease/No Change Dividend policy choices. It is evident that firms with higher ROA are likely to increase dividends rather than decreasing or maintaining them. Similar findings have been reported by Ali (2022). The results of other variables are mixed. Furthermore, COVID-19 is significantly negative in model VII, suggesting that, at this time, firms chose dividend decrease over increase. Model VIII and IX's significantly positive ROA results demonstrate that highly profitable enterprises prefer to raise their dividends rather than maintaining them.

Model X–XI describe results for Decrease vs. No-Change Dividend choice. The findings suggest that firms with higher profitability are more likely to maintain current dividend levels instead of decreasing them. Additionally, high leverage firms are more likely to decrease dividends rather than maintaining current levels.

Overall, findings show that ROA, Asset turnover, leverage, liquidity and size are the main determinants of dividend policy. Moreover, dividend omission turns out to be the preferred choice in comparison to all other dividend payout choices during COVID-19.

## 5. Conclusions

The COVID-19 pandemic had severe adverse effects on firms, investors and economies. Businesses have responded by adjusting their investing and financing policies. In this study, we analyze firm dividend payouts using annual data of 360 companies from the Pakistan Stock Exchange over the period 2015–2020.

Our analysis suggests that firms were significantly more likely to omit or reduce the amounts of dividend payments during the year 2020 when compared to the pre-COVID-19 trends in the sample period. We also observe that firm-level characteristics such as profitability, asset turnover and size affect firm dividend policies. Specifically, firms with higher profitability, asset turnover and size were less likely to opt for dividend omissions. On the contrary, dividend omissions were more likely among firms with higher debt ratios. Due to increased leverage, high-risk companies are more likely to quit paying dividends (Cejnek et al. 2021). Consequently, the market is more pessimistic during a calamity than it is under regular circumstances. Well-sized businesses prefer to raise or lower their payouts rather than eliminate them. Reducing or abolishing dividends during a crisis gives businesses more money and flexibility that may help them respond to uncertainty (Krieger et al. 2021).

Interest grew for companies that continued to make dividend payments during the pandemic (Eugster et al. 2022). To send out positive signals to the market, corporations should raise or maintain their dividend payout.

These findings have important implications for investors in designing investment strategies for crisis periods. Does the dividend payout behavior vary across industries is a question for future research, Whether the changes in cross-country dividend payouts of firms during the crisis depend on country-level factors such as inflation, culture and legal institutions might be another potential avenue for future research.

**Author Contributions:** Conceptualization, N.A. and F.S.; methodology, N.A. and F.S.; software, N.A. and F.S.; validation, B.N.A.; formal analysis, N.A. and F.S.; data curation, N.A.; writing—original draft preparation, N.A., B.N.A. and F.S.; review and editing, all authors; supervision, M.Z.U.R.; project administration, M.Z.U.R. All authors have read and agreed to the published version of the manuscript.

**Funding:** This research received no external funding.

**Informed Consent Statement:** Not applicable.

**Data Availability Statement:** Data is available upon request.

**Conflicts of Interest:** The authors declare no conflict of interest.

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
