# Peer review of "Corporate Dividend Policies during the COVID-19 Pandemic"

_economies, doi:10.3390/economies10110263_

Round 1

Reviewer 1 Report

This is an interesting paper that could be published after some minor changes have been made. Most importantly, the literature review has to be improved. In fact, there are some very interesting studies that examine the dividend policy of firms in Pakistan – in particular Ghafoor et. al. (2014) and Khan and Shamim (2017). These two papers obviously should be cited. Both studies discuss the effects of inflation on the dividend policy of companies and – at least with regard to this important question – are based on Basse (2009) and Basse and Reddemann (2011). In this context it might also be a good idea to cite Baker and Jabbouri (2016), Baker and Jabbouri (2017) and Baker, Kapoor and Jabbouri (2018). Moreover, there is a literature examining the dividend payout behavior of firms during the subprime debacle in the US and the sovereign debt crisis in Europe. Some papers that belong to this strand of the literature should also be cited.

Moreover, I would suggest to discuss one or two additional possible interesting future research questions in the conclusion (following (p. 10) “These findings have important implications for investors to design investments strategies for the crisis periods. Does the dividend payout behavior vary across industries can be the question for future research?”). In fact, it might be a good idea to focus on inflation again. It could, for instance, be quite interesting, to investigate whether the dividend payout behavior of firms in a crisis is different in countries with high and with low inflation rates.    

Literature:

Baker, H. K., & Jabbouri, I. (2016). How Moroccan managers view dividend policy. Managerial Finance, Vol. 42.

Baker, H. K., & Jabbouri, I. (2017). How Moroccan institutional investors view dividend policy. Managerial Finance, Vol. 43.

Baker, H. K., Kapoor, S., & Jabbouri, I. (2018). Institutional perspectives of dividend policy in India. Qualitative Research in Financial Markets, Vol. 10.

Basse, T. (2009). Dividend policy and inflation in Australia: results from cointegration tests. International Journal of Business and Management, Vol. 4.

Basse, T., & Reddemann, S. (2011). Inflation and the dividend policy of US firms. Managerial Finance. Vol. 37.

Ghafoor, A., Khan, M. A., Shah, S. A., & Khan, H. H. (2014). Inflation and dividend behavior of Pakistani firms: an empirical investigation using ARDL. International Journal of Business and Management, Vol. 9.

Khan, M. N., & Shamim, M. (2017). A sectoral analysis of dividend payment behavior: Evidence from Karachi Stock Exchange. SAGE Open, Vol. 7.

Author Response

Thanks for your positive evaluation of our paper. We have incorporated both of your comments.

Frist, we have updated the Introduction and literature review parts to cite your suggested literature. Specifically, we add a new paragraph at the end of Introduction section to highlight papers that have examined the dividend policies in Pakistan. This paragraphs reads as follows: 

Last but not the least we also complement the studies that investigate dividend policies of Pakistani firms (Ghafoor, Khan, Shah, & Khan, 2014; M. N. Khan & Shamim, 2017; Naimat Ullah Khan, Burton, & Power, 2011; Naimat U Khan, Jehan, & Shah, 2017; H. H. Mirza & Azfa, 2010). These studies largely examine how firm-level characteristics, such as size, leverage, profitability, cash holdings and managerial and individual ownership, and country-level factors, such as inflation and taxation, influence firm dividend decisions.  

We also add two paragraphs in Literature review to update it with new studies. These paragraphs reads as follows:

Survey based studies by H. Kent Baker and his coauthors, such as (1985), H. K. Baker and Weigand (2015), H. K. Baker and Jabbouri (2016), H. K. Baker and Jabbouri (2017) and H. K. Baker, Kapoor, and Jabbouri (2018) among others, have found that both managers and investors agree that catering, bird-in-the-hand, life cycle, signaling, and agency theories play substantial role in dividend payment decisions.

Some firm-level factors have also shown consistent relationship with firm dividend payouts. For example, Fama and French (2001) explore that firms profitable, growing and larger firms pay higher dividends. Other studies have shown that country-level factors such as inflation (Basse, 2009; Basse & Reddemann, 2011), culture (Shao, Kwok, & Guedhami, 2009; Zheng & Ashraf, 2014) and legal institutions (Ashraf, Bibi, & Zheng, 2016; Ashraf & Zheng, 2015) are significant determinants of cross-country differences in dividend policies. 

Second, we add directions for future research at the end of Conclusion part. It reads as follows:

Whether the changes in cross-country dividend payouts of firms during the crisis depend on country-level factors such as inflation, culture and legal institutions might be another potential avenue for future research?

Reviewer 2 Report

Introduction:

 - editing problems: of 3rd March 3, 2022; COVID-19 vs Covid-19; the format of the Time axis is hardly readable - better 2020-02-26

- graphs - no sense to provide two graphs to present the severity of the pandemic - one from those two. There is no sense to provide data on the graphs till 2022 while the research refers maximum till 2020.

Literature review:

 - r. 83 Dividend payments initiation or increases provide a positive signal to the market regarding a firms’ future profitability while omitting or cutting dividends provide a positive signal. - probably negative.

Data and Methodology

 r. 130 Equation - a wrong description on the right-side DIV not deltaDIV - please check at Ali (2022) if you refer this paper as a model. The definition should be precisely prepared. This is to be expected but should be stated that "i" stands for a company.

 - editing: Mkt-Bk vs MKT-Bok; Asst-Tvr vs Ast-TVR;

 Description of Table 1 is incomplete.

 - variables / models: I understand that the authors analyze pairs out of four possible behaviors of firms. This means that there should be 6 options (4 by 2 - combinations without repetition). There are only 5 options in table 1.

 - Logarithm of total assets - better clarify Natural logarithm (if such was taken).

Results and Discussions

 - This part is rather chaotic - The authors did not precisely define the individual models. Only 4 options were used in the two versions with ROA and CHE. It was not indicated whether the two other options Omission vs No-Change and Decrease vs No-Change were analyzed.

 The discussion of the results is limited to indicating which variable is significant and which is not. There are only two sentences in summary:

Overall, findings show that ROA, Asset turnover, leverage, liquidity, and size are 224 main determinants of dividend policy. Moreover, dividend omission turns out to be the 225 preferred choice in comparison to all other dividend payout choices.

 - editing: r. 167 ..... dividend omission group, Dividend Decrease .... - should be Dividend (capital letter)

Conclusion

 - The conclusion is rather limited. No suggestions or opinions were given as to why some variables influenced the decisions about Omission or Increase, etc., where these research results could be used / applied, what institutions could benefit from them.

Author Response

Introduction:

 - editing problems: of 3rd March 3, 2022; COVID-19 vs Covid-19; the format of the Time axis is hardly readable - better 2020-02-26

- graphs - no sense to provide two graphs to present the severity of the pandemic - one from those two. There is no sense to provide data on the graphs till 2022 while the research refers maximum till 2020.

Response:

We have corrected the types in dates and COVID-19. One graph is deleted and data period in graphs has been restricted to 2020. X-axis has been modified as well.

Literature review:

 - r. 83 Dividend payments initiation or increases provide a positive signal to the market regarding a firms’ future profitability while omitting or cutting dividends provide a positive signal. - probably negative.

Response:

Agree with you. It should be negative. We have revised it. 

Data and Methodology

  1. 130 Equation - a wrong description on the right-side DIV not deltaDIV - please check at Ali (2022) if you refer this paper as a model. The definition should be precisely prepared. This is to be expected but should be stated that "i" stands for a company.

 - editing: Mkt-Bk vs MKT-Bok; Asst-Tvr vs Ast-TVR;  

 Description of Table 1 is incomplete.

 - variables / models: I understand that the authors analyze pairs out of four possible behaviors of firms. This means that there should be 6 options (4 by 2 - combinations without repetition). There are only 5 options in table 1.

 - Logarithm of total assets - better clarify Natural logarithm (if such was taken).

Response:

All the points have been incorporated.

Results and Discussions

 - This part is rather chaotic - The authors did not precisely define the individual models. Only 4 options were used in the two versions with ROA and CHE. It was not indicated whether the two other options Omission vs No-Change and Decrease vs No-Change were analyzed.

 The discussion of the results is limited to indicating which variable is significant and which is not. There are only two sentences in summary:

Overall, findings show that ROA, Asset turnover, leverage, liquidity, and size are 224 main determinants of dividend policy. Moreover, dividend omission turns out to be the 225 preferred choice in comparison to all other dividend payout choices.

 - editing: r. 167 ..... dividend omission group, Dividend Decrease .... - should be Dividend (capital letter)

Response:

We have incorporated all the points by rewriting results and discussion section. Moreover, methodology section has been modified accordingly.

Conclusion

 - The conclusion is rather limited. No suggestions or opinions were given as to why some variables influenced the decisions about Omission or Increase, etc., where these research results could be used / applied, what institutions could benefit from them.

Response:

The conclusion section has been modified as well 

Reviewer 3 Report

·         In the abstract of the paper, it is necessary to highlight the purpose of the study. Also, in the abstract, describe the main methods applied (just briefly).

·         The abstract should be rewritten as to show more the new results and the author’s contribution to the science.

·         In the introduction, you stated “As of 3rd March 3, 2022, there have been 437.3 million confirmed cases and a total of 5.96 million deaths around the globe.” Could you state the data source of mentioned numbers?

·         In the introduction, you mentioned “In response, firms not only changed their day-to-day operations, such as the policies to work from home, but also financing and investing decisions including dividend payout policies.” Could you give any scientific reference that confirms this claim of yours?

·         The introduction should contain the main hypothesis of research and/or the key research questions.

·         What is the main aim of your study? Generally speaking, the paper lacks better connection and compatibility between the main aim of the paper, hypothesis, research methodology and contribution to the science.

·         At the end of the introduction, it is necessary to give a brief overview of the structure of the paper.

·         In the section 2. Literature Review, in citing and interpreting the results of other studies during the crisis, you write under the assumption that all companies made profits and dividends? However, what about companies that were operating at a loss?

·         Also, in the same section (2. Literature Review), your explanations of scientific contributions of other studies are very poor. For example, you stated “Fama and French (2001) explore that firms profitable, growing and larger firms pay higher dividends.” It is somehow expected, and can be assumed even without the results of scientific research. The key point is what is the contribution of that study to the science? Or, you stated “Cejnek et al. (2021) report that dividends substantially decreased during the first quarter of 2020 and didn’t recover until the end of the first year of COVID-19.” It is scientifically justified to mention the result of this study. But, what is the contribution of this study to the science?

·         In the section 3. Data and Methodology, you stated: “We delete financial firms from the sample because their financial ratios are not comparable to industrial firms (Xu, Lin, & Yan, 2021).” What about service companies that are not in the financial sector? Are they included in your research sample? Or, what about companies that are not industrial and belong to the real sector? Such as construction companies? Are they included in your research sample?

·         In the section 3. Data and Methodology, you stated: “We delete financial firms from the sample because their financial ratios are not comparable to industrial firms (Xu, Lin, & Yan, 2021).” However, the paper title is not completely in line with this statement and with your research sample. The same is with the explanation of scientific contributions of your study.

·         You should discuss the results obtained - how they can be interpreted in perspective of previous studies. Try to discuss the findings and their implications in the broadest context possible, including working hypothesis/research questions etc., in order to highlight new results of your paper and its new contributions.

·         In the conclusion of the paper, it is necessary to connect obtained results with the main aim of the research and the main hypothesis/research questions.

·         In the conclusion of the paper, it is needed to discuss on both theoretical and practical implications of the study.

·         What are the main limitations of your research that need to be considered when interpreting the results obtained and making conclusions?

·         In the end of the paper, references should be written according to the instructions for the authors of the journal Economies.

·         The titles of the tables in the paper are not written in the style usual in scientific papers. For example, instead of existing title of the table 1 you can write: Table 1. Description of the variables included in the model  

Author Response

We have rewritten the abstract incorporating new results and methodology.

We have added the references regarding the COVID-19 data and firm practices changes.

Our main research question is how the dividend policies of firms have changed due to the COVID-19 pandemic. This question is given at the end of 3rd paragraph of Introduction section. To examine this, we take 2015 to 2019 as normal trend in dividends and check how these normal trends in dividends changed during the year 2020 of COVID-19 outbreak.   

We have added a new last paragraph in the Introduction. It outlines the structure of the paper.   

The sample includes all non-financial companies that are included in the state bank of Pakistan's financial statement analysis (from 2015 to 2020).

We have modified the result section.

The tables has been modified accordingly.  

We have revised references in Economies style.

Round 2

Reviewer 2 Report

Please check abbreviations (still) 

 - editing: Mkt-Bk vs MKT-Bok; Asst-Tvr vs Ast-TVR;  DOM vs DO; DIC vs DI; DDC vs DD; 

There is no sens to insert the note:

ROA refers to Return on Assets, CHE is Change in earnings, Ast-TVR Asset is Turn-over, Lev is Leverage, Liq Liquidity, Size natural log of Total Assets and MKT-Bok refers to Market to Book ratio. 

under the tables 3-6. All these symbols are explained in Table 1.

Table 1: This table presents variables, their definitions, and measurements used in the study.

Author Response

Thank you so much for your comments. We have corrected the variables names as pointed out by you. We keep variables brief explanations in table notes to make it easy for readers to quickly read the tables. Hope you will conisder our point.  

Reviewer 3 Report

-

Author Response

Thanks for your review. We have rechecked the paper for language and have tried to further improve the Introduction and results sections. Hope the revised manuscript accounts for points raised by you. Thank you!